# Probabilistic Forecasting: A Level-Set Approach

**Hilaf Hasson**
Amazon Research
hashilaf@amazon.com

**Yuyang Wang**
Amazon Research
yuyawang@amazon.com

**Tim Januschowski**
Amazon Research
tjnsch@amazon.com

**Jan Gasthaus**
Amazon Research
gasthaus@amazon.com

## Abstract

Large-scale time series panels have become ubiquitous over the last years in areas such as retail, operational metrics, IoT, and medical domain (to name only a few). This has resulted in a need for forecasting techniques that effectively leverage all available data by learning across all time series in each panel. Among the desirable properties of forecasting techniques, being able to generate probabilistic predictions ranks among the top. In this paper, we therefore present Level Set Forecaster (LSF), a simple yet effective general approach to transform a point estimator into a probabilistic one. By recognizing the connection of our algorithm to random forests (RFs) and quantile regression forests (QRFs), we are able to prove consistency guarantees of our approach under mild assumptions on the underlying point estimator. As a byproduct, we prove the first consistency results for QRFs under the CART-splitting criterion. Empirical experiments show that our approach, equipped with tree-based models as the point estimator, rivals state-of-the-art deep learning models in terms of forecasting accuracy.

## 1   Introduction

Tree-based methods are known to be robust, general-purpose and high-accuracy methods for general machine learning and data science tasks, particularly those that are not image, video or text based. This is exemplified by their popularity in competitions. In a recent interview, Anthony Goldbloom, the CEO of Kaggle, called the prominence of gradient-boosted trees on Kaggle the "most glaring difference" between what is used on Kaggle and what is "fashionable in academia," a fact also reflected in the Kaggle Data Science and Machine Learning surveys [3, 4]. In time series forecasting in particular, tree-based methods have performed solidly in public competitions over the years [10], but have had a recent boost in attention via the M5 accuracy competition [23], arguably the most influential forecasting competition, which was dominated by tree-based methods. However, this recent surge in interest is not accompanied by methodological advances. In particular, while being able to generate probabilistic predictions ranks among the top desirable properties of forecasting techniques, the main tree-based method that outputs predictions for multiple quantiles remains Quantile Regression Forests (QRFs), which does not take advantage of gradient boosted trees algorithms.

In this manuscript we provide such an advancement and show its theoretical and empirical soundness: we introduce a novel algorithm, Level Set Forecaster (LSF), which can turn any point estimator algorithm into a probabilistic one. Applied to XGBoost [11] (in which case we refer to LSF as XLSF), and with a simple processing component that turns time series data into tabular data, this yields one of the first notable methodological advancements towards creating tree-based probabilistic forecasts

since the introduction of Quantile Regression Forests [27]. That said, the LSF algorithm itself is much more general: it applies to any tabular data, and it can take any point estimator algorithm.

Our contributions can be summarized as follows:

1. We propose a novel algorithm, Level Set Forecaster (LSF), which can turn any point estimator algorithm into a probabilistic one. At a high level, it groups training data whose predictions are sufficiently close, and then uses the resulting bins of true values in the training set as the predicted distributions.

2. We prove the consistency of LSF (Theorem 1); and as a byproduct, prove the consistency of QRFs (Theorem 3). By building on the methods in [32], we introduce a general framework for consistency results that goes beyond random forests, and apply it to LSF and QRF as special cases.

3. We compare LSF with the state-of-the-art models in both tabular (see Section 6.1) and forecasting tasks (See Section 6.2), and the empirical results verify the effectiveness of the proposed approach.

The rest of the article is structured as follows. In Section 2 we introduce the Level Set Forecaster algorithm, and present the consistency of the resulting estimator in Section 3. We continue by describing the method of proof of the result in Section 4, as well as presenting the novel result (Theorem 3) for QRFs. We discuss related work in Section 5, followed by the empirical studies for tabular and time series data in Section 6.

## 2 Level Set Forecaster

Given a dataset $\mathcal{D} := \{(x_i, y_i)\}_{i=1,...,n} \subset \mathbb{R}^d \times \mathbb{R}$, we train a given point prediction algorithm $\mathcal{A}$ on $\mathcal{D}$ to arrive at a model $f : \mathbb{R}^d \to \mathbb{R}$. Our goal is to output a probabilistic predictor based on $f$, such that one could query an arbitrary quantile at any testing point $x \in \mathbb{R}^d$. In more precise terms, assuming $(x_i, y_i)$'s are sampled IID from the joint distribution $(X, Y)$, we wish to find an estimator for $P(Y \leq y|X)$.

In Algorithm 1, we propose a conceptually simple yet effective approach to turn $f$ into a probabilistic forecaster. At a high level, Algorithm 1 creates groupings of training data points such that their predictions are "sufficiently close." These groupings are used to partition the feature space. With a new testing data point $x$, we find the partition cell whose predictions are "close to" $f(x)$, and the empirical samples that belong to the same partition cell yield the predictive distribution.

**Forming the partition of the feature space.** Algorithm 1 has a natural geometric interpretation. First, one trains the algorithm $\mathcal{A}$ on the data $\mathcal{D}$ to get a model $f : \mathbb{R}^d \to \mathbb{R}$. The level sets $\{f^{-1}(f(x_i))\}$ are disjoint, but they do not in general satisfy that their union covers $\mathbb{R}^d$, because it may be that $\{f(x_1), ..., f(x_n)\}$ is not equal to the entire image of $f$. We therefore create Voronoi sets for $\{f(x_1), ..., f(x_n)\}$, and look at inverse images of these. Namely, writing $\{f(x_1), ..., f(x_n)\}$ as $\{v_1, ..., v_k\}$ without repetition and with $v_1 < \cdots < v_k$, the $j^{th}$ Voronoi set is:

$$V_j = \{a \in \mathbb{R}|v_j = \text{argmin}_{\{v_1,...,v_k\}}(|v_i - a|)\}. \tag{1}$$

We now define $\tilde{B}_j = f^{-1}(V_j)$, with each $\tilde{B}_j$ being a disjoint union of level sets of $f$. The inverse Voronoi sets $\tilde{B}_j$'s are disjoint, and their union covers all of $\mathbb{R}^d$. This is not yet, however, our final partition: with the basic set-up described above, the cardinality of each bin $|\{y_i \in \{y_1, ..., y_n\}|x_i \in \tilde{B}_j\}|$ may be too small (often equal to 1) to ensure the desired property of consistency. In the final part of the algorithm we merge the different $\tilde{B}_j$ together, by going sequentially over $\tilde{B}_1, ..., \tilde{B}_k$ (associated with the ascending values $v_1, ..., v_k$), grouping them until a sufficient bin size is ensured (repeated $y_i$ values are not ignored in this computation), and then proceeding to a new grouping, arriving at the final partition $\mathbb{R}^d = \bigcup B_j$, where each $B_j$ is a union of $\tilde{B}_i$'s. The intuition behind the partitioning algorithm is that for a test sample $(X, Y)$ and a value $y \in \mathbb{R}$ we expect $P(Y \leq y|X)$ to not vary wildly as $X$ runs over a particular partition cell $B_j$.

---

**Algorithm 1:** Level Set Partitioning Algorithm

---

**Input**: $\mathcal{D} := \{(x_i, y_i)\}_{i=1,...,n} \subset \mathbb{R}^d \times \mathbb{R}$, a natural number `min_bin_size`, a point prediction algorithm $\mathcal{A}$ (e.g., XGBoost).

Train $\mathcal{A}$ on $\mathcal{D}$, and call the resulting model $f : \mathbb{R}^d \to \mathbb{R}$.

Create a dictionary `pred_to_bin` whose keys are $\{f(x_i)\}_i$, such that the value of $f(x_i)$ is the list of $y_j$ satisfying $f(x_j) = f(x_i)$.

Sort the distinct elements of $\{f(x_1), ..., f(x_n)\}$ in ascending order, and denote them $[v_1, ..., v_k]$.

Initialize an empty dictionary `res_pred_to_bin`, and an empty list `current_bin`.

*# If the size of a bin is smaller than the minimum required size, successively merge it with the bins on its right (higher value) until the minimum size constraint is satisfied.*

**for** $v$ in $[v_1, ..., v_k]$ **do**
  Concatenate `pred_to_bin[v]` at the end of `current_bin`.
  `res_pred_to_bin[v]` ← `current_bin` (pass `current_bin` by reference)
  **if** `len(current_bin)` ≥ `min_bin_size` **then**
    ⌊ Point `current_bin` to a new empty list object.

*# If the last bin is too small, merge to the left.*

**if** `len(res_pred_to_bin[`$v_k$`])` < `min_bin_size` **then**
  concatenate `res_pred_to_bin[`$v_i$`]` to `res_pred_to_bin[`$v_k$`]`, where $i$ is the maximal index satisfying that `res_pred_to_bin[`$v_i$`]` $\neq$ `res_pred_to_bin[`$v_k$`]`.

`pred_to_bin` ← `res_pred_to_bin`

---

**Output**: A partition such that $w_1, w_2 \in \mathbb{R}^d$ are said to be in the same partition if `pred_to_bin` maps $\arg\min_{\{v_1,...,v_k\}}(|v_i - f(w_1)|)$ and $\arg\min_{\{v_1,...,v_k\}}(|v_i - f(w_2)|)$ to the same bin.

---

**Using the partition to make inferences.** In the notation of Algorithm 1, the algorithm finds the closest element $v_j$ in $[v_1, ..., v_k]$ to $f(x)$ (by binary search), and then computes and caches the quantile of `pred_to_bin[`$v_j$`]`.

In more mathematical terms, Algorithm 1 outputs a partition $\mathbb{R}^d = \bigsqcup B_j$ of the feature space into finitely many disjoint sets. Queried at a feature vector $x \in B_l \subset \mathbb{R}^d$, and for any value $y \in \mathbb{R}$, our probabilistic forecasting algorithm returns an estimator for $P(Y \leq y|X)$ at $X = x$:

$$\hat{\eta}_{LSF}(x) := \frac{|\{\gamma \in L_l | \gamma \leq y\}|}{|L_l|}, \tag{2}$$

where $L_l := \{y_i \in \{y_1, ..., y_n\}|x_i \in B_l\}$ is the associated true target values for the partition cell $B_l$. We shall study the consistency property of the estimator $\hat{\eta}_{LSF}$ in the next section.

**Remark 1.** Algorithm 1 resembles the classic Quantile Regression Forecasts (QRFs) ([27]) in the following sense. In QRFs, the random forests are trained in the regular sense, i.e., one splits the nodes according to the CART-splitting criterion. The probabilistic predictions are generated by "opening" the leaf nodes of each tree, and empirically sampling the predictions instead of predicting with the average values in the leaves. Here, the second step is exactly the same while the "leaf nodes" are replaced by partitions induced by the level sets of an arbitrary point estimator. In fact, if one used a decision tree for $\mathcal{A}$ and let `min_bin_size` be 1 then Algorithm 1 reduces to a QRF with one tree.

We refer to the estimator described via Equation 2 with the partitioning algorithm as in Algorithm 1 as the Level Set Forecaster (LSF)[1] associated to $\mathcal{A}$. If $\mathcal{A}$ is XGBoost, we refer to it as XLSF.

## 3   Main Theorem: LSF is Consistent

In this section, we aim to provide a quantitative understanding of the Level-set Forecaster (LSF), in particular, the consistency of LSF. The key insight that motivated the LSF algorithm was the observation that the literature regarding the consistency of Random Forests (RFs) can be extended to apply in much more general contexts.

---

[1]LSF is implemented in GluonTS: https://github.com/awslabs/gluon-ts/blob/master/src/gluonts/model/rotbaum/README_LSF.ipynb

We remark that it is precisely because the partitions of trees grown by the CART-splitting algorithm are data-dependent that consistency of RFs has only been proven recently [32], while most other work ([7, 36, 6, 27, 31]) has focused on easier algorithms that are not used in practice. We can leverage these methods to prove consistency of LSF as well (Theorem 1); and as a byproduct we also prove consistency of QRFs under the CART-splitting algorithm (Theorem 3).

In order to have a consistency assurance, we require mild assumptions on the data generating process, and some intuitive assumptions on the point forecasting algorithm $\mathcal{A}$.

**Assumption 1.**

1. ***Assumption on the data generating process:***

   (a) *There is a uniformly equicontinuous family of functions $p_y(x)$ (as $y$ varies) so that $p_y(x)$ integrates to $P(Y \leq y|X)$ for all $y$. Further assume that $\mathbb{E}(Y|X)$ is bounded.*

   (b) *For every $\varepsilon > 0$ there is a $\delta > 0$ so that for every $x$ in the image of $X$ the probability that $X$ lies in the ball of radius $\varepsilon$ centered at $x$ is at least $\delta$.*

   (c) *There exists a function $m(x)$ that integrates to $\mathbb{E}(Y|X)$ so that $\forall \varepsilon > 0 \, \exists \delta > 0$ such that for all $x, x' \in \mathbb{R}^d$ and $0 \leq y \leq 1$: $|m(x) - m(x')| < \delta$ implies $|p_y(x) - p_y(x')| < \varepsilon$.*

2. ***Assumptions on the point estimator*** $\mathcal{A}$: *Let $W_n := \{f(X_1), ..., f(X_n)\}$, with $k := |W_n|$. Assume that $\mathcal{A}$ has a choice (for every $n$) of hyperparameters that would satisfy:*

   (a) *Collisions among the training data $f(X_i) = f(X_j)$ are rare in the precise sense that there exists a positive number $C$ such that:*

   $$P\left(C < \frac{k}{n}\right) \to 1.$$

   (b) *For $X$ an independent variable following the distribution of the $X_i$'s the value $f(X)$ does not tend to be extremal among the $f(X_i)$'s. To be precise, for any sequence $d_n \to 0$ of positive numbers we have that*

   $$P\left(\frac{|\{v \in W_n | f(X) < v\}|}{k} > d_n\right) \to 1, P\left(\frac{|\{v \in W_n | v < f(X)\}|}{k} > d_n\right) \to 1.$$

   (c) *The image of $f$ is "dense in probability". To be precise,*

   $$(\ln(n))^2 \max_{i=2,...,k-1} \text{Vol}(V_i) \to 0$$

   *in probability, where the $V_i$'s are the Voronoi sets from Equation 1.*

   (d) *There's a constant $c$ so that $\forall j, \forall x \in f^{-1}(V_j) \forall \varepsilon > 0$ the set $f^{-1}(V_j) \cap B_\varepsilon(x)$ a.s. contains a ball of radius $c \cdot \varepsilon$.*

   (e) *The base algorithm $\mathcal{A}$ is a mean square consistent estimator of the conditional mean:*

   $$\mathbb{E}(|f(X) - \mathbb{E}(Y|X)|^2) \to 0.$$

**Theorem 1.** *Under Assumption 1, letting* `min_bin_size`$= (\ln(n))^2$, *LSF is mean square consistent. That is, for any value $y \in \mathbb{R}$, we have*

$$\mathbb{E}(|\hat{\eta}_{LSF}(X) - P(Y \leq y|X)|^2) \to 0, \tag{3}$$

*where the convergence is uniform in $y$.*

As a byproduct of proving this result, we also give the first consistency result (Theorem 3) for QRFs grown under the CART-splitting algorithm under an additive regression model assumption.

**Remark 2.** Assumptions 1a and 1b are much weaker than the additive regression assumptions in [32]. Assumption 1c is what allows the base algorithm to be informative about the conditional pdfs. In light of the empirical success of LSF (see Section 6), it appears that Assumption 1c holds quite generally in real-world data. We expect that Assumptions 2a, 2b, 2c, and 2d, which simply ensure non-degenerate behavior, hold for any reasonable choice of $\mathcal{A}$, but the precise statement is left as a future work. (If $f$ is locally constant on hyperrectangles then Assumption 2d holds with $c = \frac{1}{\sqrt{d}+1}$.)

# 4 Method of Proof: Extend Results on RF Consistency to More General Settings

Our strategy for proving Theorem 1 is to generalize and strengthen existing results on the consistency of RFs, in particular through the work of [32]. We propose a unified framework that subsumes both the standard RFs (and QRFs) and the proposed LSF estimator, allowing us to arrive at the consistency results. We begin by introducing additional notation and nomenclature.

## 4.1 Problem Setup and Existing Results

Define a *data-based partitioning algorithm* $\mathcal{B}$ as being any algorithm taking a training dataset, i.e., a finite subset $\mathcal{D} = \{(X_1, Y_1), ..., (X_n, Y_n)\}$ of $\mathbb{R}^d \times \mathbb{R}$ and outputting a (potentially random) partition[2] of $\mathbb{R}^d$ into disjoint sets $\bigsqcup B_j$ satisfying that there's a constant $c > 0$ so that $\forall j \forall x \in B_j \forall \varepsilon > 0$ the set $B_j \cap B_\varepsilon(x)$ a.s. contains a ball of radius $c\varepsilon$. (This holds if the $B_j$'s are hyperrectangles using $c = \frac{1}{\sqrt{d}+1}$.) Examples include Algorithm 1, as well as any tree-growing algorithm such as the CART-splitting algorithm (see Algorithm 2, recalled below).

We define a mean-regression (resp. quantile-regression) $\mathcal{B}$-estimator, trained on $\mathcal{D}$ as an estimator of $\mathbb{E}(\xi(Y)|X)$, where $\xi$ is the identity map $\xi(Y) := Y$ (resp. $\xi(Y) := I_{Y \leq y}$) given by first applying $\mathcal{B}$ to get a partition $\mathbb{R}^d = \bigsqcup B_j$; and then at inference taking a feature vector $x \in B_l$ to: $\frac{\sum_{\gamma \in L_l} \xi(\gamma)}{|L_l|}$, where $L_l := \{Y_i \in \{Y_1, ..., Y_n\} | X_i \in B_l\}$. Now we are ready to introduce the central object of our study, *generalized RF/QRF*,

**Definition 1.** *A Generalized RF (resp. Generalized QRF) grown using the partitioning algorithm $\mathcal{B}$ with $M$ estimators is defined as:*

$$\hat{\eta}(x) := \frac{\sum_{j=1}^M \hat{\eta}_j(x)}{M}, \tag{4}$$

*where $\hat{\eta}_j$ is a mean-regression (resp. quantile-regression) $\mathcal{B}$-estimator trained on only part of the data: for the $j^{th}$ estimator subsample uniformly without replacement $a_n$ many datapoints from $\mathcal{D}$ to train on, where $a_n$ is a (non-random) sequence of natural numbers satisfying $1 \leq a_n \leq n$.*

We remark that LSF is simply a generalized QRF grown using Algorithm 1 with one estimator ($M = 1$) and full subsampling ($a_n = n$); and that RFs (resp. QRFs) are simply generalized RFs whose partitioning algorithm is induced by a tree-growing algorithm such as the CART splitting algorithm (see Algorithm 2).

To fix some notation, we let $t_n$ (a random variable that can depend on the data) be the number of partitions in the first $\mathcal{B}$-estimator, and we let $R_{x,n}$ be the partition in the first $\mathcal{B}$-estimator that contains a feature vector $x \in \mathbb{R}^d$.

The main question we consider is:

**Problem 1.** *Under what assumptions is the generalized RF/QRF estimator $\hat{\eta}(\cdot)$ constructed in (4) $L^2$ (mean square) consistent? i.e.,*

$$\mathbb{E}(|\hat{\eta}(X) - \mathbb{E}(\xi(Y)|X)|^2) \to 0,$$

*where $\xi$ is defined either as $\xi(T) = T$ or $\xi(T) = I_{T \leq y}$.*

Our investigation starts with the recent work on the consistency of random forests. In Theorem 1 of [32] the authors proved mean square consistency in the mean regression case ($\xi(T) = T$) under some mild assumptions, where the partitioning algorithm is the one induced by a tree grown via the CART splitting criterion.

We briefly recall the tree-growing algorithm for the CART splitting criterion:

---

**Algorithm 2:** Mean and Quantile Regression Forests with the CART-Splitting Criterion; CART loss is recalled in Appendix A. Note that unlike Algorithm 1 the number of partition cells $t_n$ here is a hyper-parameter and therefore not random.

---

**Input**: We fix a natural number $m_{try} \leq d$; and sequences of natural numbers $\{t_n\}_n$, $\{a_n\}_n$ such that $t_n \leq a_n \leq n$.;

**while** *not reaching $t_n$ many leaves* **do**

> For the subsampling of each tree, choose $a_n$ points uniformly without replacement from the training set.
>
> At each step, and for each leaf, choose the split that minimizes CART loss, considering only $m_{try}$ split dimensions sampled uniformly without replacement from $\{1, ..., d\}$.

---

**Mean prediction**: output the average value of the leaf nodes. ($\xi(T) = T$);
**Quantile prediction**: output the average value of the leaf nodes. ($\xi(T) = I_{T \leq y}$)

---

In [32], the authors make the following assumptions:

**Assumption 2** (Assumptions in [32]).

1. *Subsamples and tree growth are same as in the CART-splitting algorithm.*

2. *Additive Regression Model Assumption: $Y$ is normal with fixed variance $\sigma^2$ and with mean $\sum_{i=1}^{d} m_i(p_i(X))$, where $p_i$ is the projection onto the $i^{th}$ coordinate, where each $m_j : [0,1] \to \mathbb{R}$ is a continuous function, and where the marginal distribution of $X$ is uniform in the hypercube $[0,1]^d$.*

3. *Assume that $\lim_{n \to \infty} \dfrac{t_n \ln(a_n)^9}{a_n} = 0, \lim_{n \to \infty} t_n = \infty$.*

The result of primary interest in [32] is the following.

**Theorem 2.** *([32], Theorem 1) Under Assumptions 2 we have mean square consistency*

$$\mathbb{E}\left[ |\hat{\eta}_1(X) - \mathbb{E}(Y|X)|^2 \right] \to 0,$$

*where $\xi(T) := T$ is the identity transformation.*

**Remark 3.** The authors [32] phrase their result as the consistency of $\mathbb{E}(\hat{\eta}_1(X)|X, \{X_i, Y_i\}_i)$, i.e. the case of an infinite number of trees, but they in fact reduce first to the stronger statement that a single tree is consistent.

In [27], the quantile regression case ($\xi(T) = I_{T \leq y}$) was addressed, but only for the case of label-independent weights, which precludes the case of the partitioning algorithm being induced by trees grown via the CART-splitting algorithm.

## 4.2 Consistency of Generalized RFs/QRFs

To begin with, we state a novel result, the first consistency proof for QRFs under the CART-splitting criterion.

**Theorem 3.** *Under Assumption 2, with the assumption 2.3 replaced by the weaker assumption that $\lim_{n \to \infty} \dfrac{t_n \ln(a_n)}{a_n} = 0, \lim_{n \to \infty} t_n = \infty$, we have that:*

$$\mathbb{E}(|\hat{\eta}_1(X) - P(Y \leq y|X)|^2) \to 0,$$

*uniformly in $y$, where $\xi(T) := I_{T \leq y}$.*

We remark that convergence in probability (the notion used in [27]) and $L^2$ convergence coincide for QRFs because the estimator and the CDF are both bounded.

Both Theorems 3 and Theorem 1 are corollaries from the following generalization of Theorem 2:

---

[2]In order to be rigorous one really ought to use random closed sets. We will ignore this subtlety for convenience, and assume that the clever reader can deduce the proper adjustments.

**Theorem 4.** *A generalized QRF with a single estimator (a "generalized quantile regression tree") is mean square consistent under the data generating assumptions 1a, 1b in Assumption 1, and the following conditions:*

1. *The expected variance of the conditional CDFs in the cell containing $X$ goes to $0$ uniformly in the quantile. To be more precise, for $(X', Y')$ an IID copy of $(X, Y)$ restricted to the sub probability space where $X' \in R_{X,n}$, assume that $\mathbb{E}(V(P(Y' \leq y|X')|X))$ goes to $0$ uniformly in $y$.*

2. $\dfrac{t_n \ln(a_n)}{a_n} \to 0$ *in probability.*

*Namely, under these conditions, we have that:*

$$\mathbb{E}\left[|\hat{\eta}_1(X) - P(Y \leq y|X)|^2\right] \to 0,$$

*where the convergence is uniform in $y$, and where $\xi(T) := I_{T \leq y}$.*

Theorem 1 follows from Theorem 4 as an immediate corollary:

*Proof.* (Theorem 1) By Assumption 2a, and our choice `min_bin_size`$= (\ln(n))^2$, it follows that in probability $t_n$ has the same order of magnitude as $\frac{n}{(\ln(n))^2}$. It is therefore clear that Assumption 2 in Theorem 4 is satisfied.

Note that $R_{X,n}$ is the union of at most $(\ln(n))^2$ many $f^{-1}(V_j)$. By Assumption 2b, with probability converging to $1$ none of these $V_j$'s are $V_1$ nor $V_{t_n}$. Now Assumption 2c immediately implies that $\text{Vol}(f(R_{X,n}))$ goes to $0$ in probability. Letting $(X', Y')$ be an IID copy of $(X, Y)$ restricted to the sub probability space where $X' \in R_{X,n}$, this implies that $\mathbb{E}(V(f(X')|X)) \to 0$. By Assumption 2e, we have that $V(f(X) - \mathbb{E}(Y|X)) \to 0$, and thus (by the law of total variance) $\mathbb{E}(V(f(X') - \mathbb{E}(Y'|X')|X)) \to 0$. Therefore $\mathbb{E}(V(\mathbb{E}(Y'|X')|X)) \to 0$. By an application of Chebyshev's Inequality and the boundedness of $\mathbb{E}(Y|X)$, Assumption 1c implies that $\mathbb{E}(V(P(Y' \leq y|X')|X)) \to 0$ uniformly in $y$.

$\square$

Note that Assumptions 2a, 2b and 2c could have been replaced simply by $\mathbb{E}(V(f(X')|X))$ going to $0$ in the notation of the proof above. It is less obvious to see why Theorem 4 implies Theorem 3. The assertion boils down to proving that the assumptions of Theorem 3 imply Assumption 1 of Theorem 4. We refer to Appendix B for the proof of Theorems 3 and 4.

## 5   Related Work

The classical conformal prediction algorithm [33], whose underlying principle is bootstrapping the residuals on a validation set, is another approach for turning point estimator algorithms into probabilistic ones. The natural setting in this approach is to create prediction intervals, rather than to make predictions for particular quantiles. Unlike LSF, the prediction intervals from conformal predictions are always symmetric about the point prediction with a fixed length. More sophisticated conformal prediction algorithms exist that allow for variable length prediction intervals (e.g. [29], which takes as input two quantile regression estimators), but they no longer satisfy that they take a single point estimator algorithm to a probabilistic one. We refer to Appendix C for more details.

The main available probabilistic tree-based algorithm other than LSF (applied to a tree-based point estimator algorithm) that does not require a list of quantiles during training is QRFs [27]. We remark that LSF applied to a decision tree is equivalent to a QRF with a single tree, and that therefore QRFs with multiple trees are an ensemble of these simple LSF algorithms.

Another approach for creating tree-based probabilistic predictions is to learn the parameters of a family of conditional distributions (e.g., [9]). For tree-based methods, this has been proposed [25, 26]. We do not further compare against this method in our experiments because the implementations are not available and instead focus on comparisons with state-of-the-art methods for forecasting that use a similar approach (e.g., DeepAR [30]).

For probabilistic time series forecasting, a number of global [18] models have been proposed (e.g., [28, 12, 22, 14, 37]). For a comprehensive review for neural network models for forecasting, please refer to [16, 5]. In contrast to classical forecasting models (e.g., [17]) which are local (parameters are estimated per time series independently), tree-based methods are best employed as global models (learning parameters over the entire panel of time series, see e.g., [19] for a discussion and [35] for a concurrent contribution). In our empirical comparisons, we chose DeepAR [30] and CNN-QR [38] as global, deep-learning based models for their robust state-of-the-art performance.

## 6 Experiments

In Section 6.1 we compare XLSF (LSF with XGBoost) against QRFs (as a leading tree-based probabilistic algorithm) and Conformalized Predictions (as a baseline for turning point estimators into probabilistic ones). In Section 6.2 we apply LSF to time series data. All experiments are done using Amazon Sagemaker [21] with instance type `ml.m4.16xlarge`. All hyperparameters used are specified in Appendix F.

### 6.1 Experiments on Tabular Data

A main competitor for LSF for using tree-based methods to create probabilistic predictions is QRFs ([27]). In the experiments we use the skgarden implementation ([2]). QRFs are made up of multiple trees, but XLSF is one generalized tree made up of an XGBoost ensemble. It is therefore interesting to see how the two compare. By design, conformal predictions are phrased in terms of prediction intervals (a pair of quantiles) rather than a range of quantiles. For example for an $\alpha = 0.1$, conformal predictions aim to estimate P05 and P95 so as to have an accuracy of the prediction intervals (percent of times that the true value is in the prediction interval) that is at least 90%. We therefore chose only the pair of quantiles P05 and P95 in our experiments, as opposed to 3 quantiles as in Section 6.2.

The results are shown in Table 1, and we see that XLSF outperforms conformal predictions by a wide margin on quantile prediction, though conformal predictions' prediction interval accuracy was more faithfully close to 90%. We remark that to reap the benefits of both algorithms, one can feed XLSF into the "Conformalized Quantile Prediction" algorithm; see [29], and the discussion in Appendix C. We also remark that QRF is much slower compared to XLSF, as shown in Table 1. The datasets in this section were small, but in Section 6.2 we will see that this makes classical QRF impractical for time series predictions, while XLSF is quite competitive.

| | XLSF | | | | QRF | | | | Conformalized Predictions | | | |
|---|---|---|---|---|---|---|---|---|---|---|---|---|
| | P05 | P95 | accuracy | time (s) | P05 | P95 | accuracy | time (s) | P05 | P95 | accuracy | time (s) |
| facebook$_1$ | 0.103 | **0.288** | 94.31% | 7.668 | **0.094** | 0.317 | 92.74% | 29.918 | 0.471 | 0.428 | 89.98% | 6.942 |
| facebook$_2$ | 0.097 | **0.293** | 95.32% | 14.790 | **0.094** | 0.299 | 92.93% | 103.61 | 0.391 | 0.445 | 89.89% | 14.442 |
| meps$_{19}$ | **0.100** | **0.562** | 93.37% | 5.874 | 0.111 | 0.687 | 89.45% | 9.159 | 0.633 | 0.766 | 89.64% | 5.305 |
| meps$_{20}$ | **0.100** | **0.664** | 92.78% | 6.384 | 0.107 | 0.674 | 88.37% | 10.526 | 0.498 | 0.774 | 90.13% | 5.884 |
| meps$_{21}$ | **0.100** | **0.555** | 92.68% | 6.215 | 0.105 | 0.635 | 88.40% | 9.111 | 0.525 | 0.802 | 89.94% | 5.246 |
| concrete | 0.036 | 0.039 | 76.69% | 0.627 | 0.036 | **0.037** | 82.52% | 0.167 | **0.031** | 0.042 | 86.89% | 0.089 |
| star | **0.011** | 0.012 | 78.52% | 1.004 | **0.011** | 0.014 | 79.21% | 0.500 | **0.011** | **0.011** | 87.99% | 0.305 |
| bio | 0.082 | 0.132 | 87.63% | 13.195 | **0.073** | **0.096** | 84.40% | 33.153 | 0.103 | 0.128 | 89.99% | 4.528 |
| community | 0.105 | 0.189 | 76.19% | 1.120 | **0.079** | 0.187 | 86.71% | 1.132 | 0.136 | **0.184** | 92.48% | 0.955 |
| bike | 0.056 | 0.059 | 87.74% | 3.403 | **0.043** | **0.044** | 80.53% | 2.828 | 0.050 | 0.048 | 91.23% | 0.734 |

Table 1: Benchmarking results: the datasets were taken from https://github.com/yromano/cqr/tree/master/datasets. Accuracy is percent of times the true value was in the prediction interval. (It should revolve around 90%.) P05 and P95 are weighted quantile losses.

### 6.2 LSF for Time Series Prediction

In order to apply LSF to time series data, a few choices need to be made. First, the method for preprocessing the data into tabular data; second, whether to use a single model for the entire forecast

| quantile | LSF-wrapping of winning M5 solution | Winner | Runner-up | Third | Fourth | Fifth |
|---|---|---|---|---|---|---|
| 0.005 | 0.01227 | 0.010 | 0.04201 | 0.01891 | 0.01225 | 0.01133 |
| 0.025 | 0.05484 | 0.05004 | 0.08649 | 0.06446 | 0.05483 | 0.05739 |
| 0.165 | 0.31677 | 0.31276 | 0.33782 | 0.33371 | 0.31566 | 0.35329 |
| 0.25 | 0.45108 | 0.44436 | 0.46158 | 0.46699 | 0.44937 | 0.49415 |
| 0.5 | 0.72434 | 0.69886 | 0.69044 | 0.74712 | 0.71279 | 0.7661 |
| 0.75 | 0.72848 | 0.68747 | 0.72231 | 0.69522 | 0.71071 | 0.70321 |
| 0.835 | 0.60748 | 0.58519 | 0.59897 | 0.59059 | 0.61292 | 0.59058 |
| 0.975 | 0.21801 | 0.18236 | 0.19664 | 0.19268 | 0.19348 | 0.20365 |
| 0.995 | 0.0863 | 0.0551 | 0.07689 | 0.07433 | 0.06225 | 0.06811 |
| mean_wQL | 0.35551 | **0.33624** | 0.35702 | 0.35378 | 0.34714 | 0.36087 |

Table 2: The LSF-wrapping of the winning M5 accuracy competition's solution on the top point forecast submission compared with top results in the uncertainty competition. Note that the non-monotonous nature of the metrics for the winning solutions is due to the fact that all of the numbers in this table are being evaluated only on the bottom level of the hierarchy, and that the metric used is different from the competitions' metric.

horizon, or one model for each timestep; and finally, one has to decide on what point estimator algorithm $\mathcal{A}$ to use and its hyperparameters. The winning solution to the M5 accuracy competition has already made all of these decisions, and so we can simply wrap it with LSF, and see how it compares with the top 5 solutions for the M5 uncertainty competition. We also propose a simple default choice (using only lag features, one model for each timestep, and using XGBoost) and apply it to multiple datasets.

**M5 competition** The M-competitions are a series of competitions led by Spyros Makridakis intended to evaluate and compare the accuracy of different forecasting methods. The most recent competition, the M5 competition [23], was in fact a twin competitions: an accuracy competition (point forecasting) and an uncertainty competition (probabilistic forecasting) with the same training data. The winning solution (by YeonJun-IN, available in [1]) to the M5 accuracy competition is a tree-based solution that trains 220 lightgbm ([20]) models (with different feature engineering set ups and/or trained on different parts of the data) and combines them to create a point forecast. The preprocessing involved more than mere lag features.

We remark that the M5 uncertainty competition requires forecasts not only of the individual time series, but also of hierarchical aggregates. In our reported metrics we only consider the bottom level of the hierarchy, since there is no trivial way to produce aggregate forecasts using the LSF-wrapping of the winning M5 accuracy competition's solution.

We can see that wrapping the M5 accuracy competition's winning solution is competitive among the top 5 solutions for the M5 uncertainty competition, showing the out-of-the-box we were able to turn quality point-forecasts into quality probabilistic forecasts.

**Benchmarking datasets.** We now turn to some benchmarking with certain default choices of preprocessing and inference on common time series datasets. To be precise, we preprocess the time series data into tabular data by sampling $P$ context windows with only lag features, and then training a separate probabilistic forecaster for each time-step in the future. (See Algorithm 3 in the appendix for more details; see also Appendix F.)

We refer to this set-up as Tree-based Time Series Wrapper (TTSW). We remark that by design, TTSW with XLSF never predicts above (resp. below) the maximal (resp. minimal) value observed. We also include TTSW with QRFs instead of XLSF, and with lightgbm ([20]) with quantile loss, which we refer to as Quantile Regression. If one chooses XLSF or QRF there is no need to specify quantiles at training time, but for Quantile Regression, one specifies quantiles before training, and for each timestep in the forecast horizon it trains as many models as the number of quantiles requested.

|  | TTSW (XLSF) | TTSW (QR) | TTSW (QRF) | DeepAR | CNN-QR |
|---|---|---|---|---|---|
| electricity [13] | **0.0499** | 0.1478 | 0.0785 | 0.0592 | 0.0601 |
| parts [34] | 0.9709 | 0.7584 | 0.9617 | 0.9877 | **0.7566** |
| m4_daily [24] | **0.0148** | 0.0281 | 0.0188 | 0.0239 | 0.0154 |
| traffic [13] | 0.1212 | 0.1062 | 0.1289 | **0.0913** | 0.1134 |
| wiki10k | 0.2926 | 0.2534 | 0.3069 | **0.2421** | 0.2482 |
| dcrideshare [8] | 0.3381 | 0.3171 | 0.3929 | 0.2837 | **0.2820** |

Table 3: Mean Weighted Quantile Loss (across P10, P50, P90).

By default $P = 1000000$ and the context window is equal to the forecast horizon. In Table 3, since QRF is much slower to train, we had to reduce $P$ to 10000, which means it had 100 times less data-points to train on. Even with this adjustment, it took considerably longer than the other methods. Our results in Table 3 show that TTSW (both XLSF and with Quantile Regression) perform competitively with state-of-the-art deep learning forecasting methods ([30, 38]), while maintaining advantages of tree-based methods such as interpretability. For detailed results, please refer to Appendix E.

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
