# A CART Loss

Let $\tau_{i,j}$ be the indicator variable for whether $X_i$ is chosen for the $j^{th}$ estimator. We recall that CART loss for the a cell $A$ along dimension $k$ for the position cut $z$ in tree $j$ is:

$$\frac{1}{\sum_i I_{X_i \in A} \tau_{i,j}} \sum_i (Y_i - \bar{Y}_A)^2 I_{X_i \in A} \tau_{i,j} - \frac{1}{\sum I_{X_i \in A} \tau_{i,j}} \sum_i (Y_i - \bar{Y}_{A_L} I_{X_i^{(k)} < z} - \bar{Y}_{A_R} I_{X_i^{(k)} \geq z})^2 I_{X_i \in A} \tau_{i,j},$$
(5)

where the superscript $^{(k)}$ denotes the $k$-th coordinate, and where

$$\bar{Y}_A = \frac{\sum Y_i I_{X_i \in A} \tau_{i,j}}{\sum I_{X_i \in A} \tau_{i,j}}, \quad \bar{Y}_{A_L} = \frac{\sum Y_i I_{X_i \in A} I_{X_i^{(k)} < z} \tau_{i,j}}{\sum I_{X_i \in A} I_{X_i^{(k)} < z} \tau_{i,j}}, \quad \text{and} \quad \bar{Y}_{A_R} = \frac{\sum Y_i I_{X_i \in A} I_{X_i^{(k)} \geq z} \tau_{i,j}}{\sum I_{X_i \in A} I_{X_i^{(k)} \geq z} \tau_{i,j}}.$$

If any of the denominators are 0, by convention, the CART loss is $\frac{1}{\sum_i I_{X_i \in A} \tau_{i,j}} \sum_i (Y_i - \bar{Y}_A)^2 I_{X_i \in A} \tau_{i,j}$.

# B Proofs

## B.1 Proof of Theorem 3

Recall that cuts under the CART splitting criterion are the cuts given by Algorithm 2, which uses the CART loss given by Equation 5. In [32] theoretical cuts are defined as the cuts that follow Algorithm 2, but with the CART loss replaced by a "theoretical loss". The theoretical loss, in turn, is defined for a cell $A$ along dimension $k$ for the position cut $z$ as:

$$L^*(k, z) := V(Y | X \in A) - P(X^{(k)} < z | X \in A) V(Y | X^{(k)} < z, X \in A)$$
$$- P(X^{(k)} \geq z | X \in A) V(Y | X^{(k)} \geq z, X \in A)$$

The reason for the name "theoretical cuts" is that by the strong law of large numbers CART loss converges almost surely to the theoretical loss, which is not dependent on the data.

We first remark that in [32] their proof of Lemma 1 in fact proves more than its statement:

**Lemma 1.** *Under Assumption 2.2, let $x \in \mathbb{R}^d$, let $I_k$ be the hyperrectangle containing $x$ after $k$ theoretical cuts, and let $D_x := \cap_k I_k$. Then with probability 1 we have that $\mathbb{E}(Y|X)$ is almost surely constant over $D_x$.*

*Proof.* This is immediate from the proof of Lemma 1 in [32]. (Note that the randomness of $D_x$ in the statement comes from the choice of splitting dimensions, governed by $\theta$.) □

As Assumptions 1a, 1b and 1c are trivially satisfied under the additive regression model assumption, similarly to the proof of Theorem 1 it suffices to show that $\mathbb{E}(V(\mathbb{E}(Y'|X')|X)) \to 0$, where $(X', Y')$ is an IID copy of $(X, Y)$ restricted to the sub probability space where $X' \in R_{X,n}$. This is an immediate corollary of the following lemma:

**Lemma 2.** *Under Assumption 2.2, let $D_X := \cap_k I_k$, where $I_k$ is the hyperrectangle containing $X$ after $k$ theoretical cuts. Then the volume $\mathrm{Vol}(R_{X,n} \backslash D_X)$ converges to 0 in probability.*

*Proof.* Fix $\varepsilon, \delta > 0$. As in [32], let $A_{k,n}(X, \theta)$ be the cell containing $X$ after only $k$ cuts, and let $A_k^*(X, \theta)$ be the cell containing $X$ after $k$ theoretical cuts. By definition, there exists a $k_0$ so that $P(\mathrm{Vol}(A_{k_0}^*(X, \theta) \backslash D_X) > \frac{\varepsilon}{2}) < \frac{\delta}{2}$. By Lemma 3 in [32], the cuts up to the $k_0^{th}$ step are arbitrarily close to the theoretical cuts with as high a probability as we want. In particular, for any $n$ big enough, we have that

$$P(\mathrm{Vol}(A_{k_0,n}(X, \theta) \backslash A_{k_0}^*(X, \theta)) > \frac{\varepsilon}{2}) < \frac{\delta}{2}.$$

As one can observe from a simple Venn diagram,

$$\mathrm{Vol}(A_{k_0,n}(X, \theta) \backslash D_X) \leq \mathrm{Vol}(A_{k_0,n}(X, \theta) \backslash A_{k_0}^*(X, \theta)) + \mathrm{Vol}(A_{k_0}^*(X, \theta) \backslash D_X).$$

Therefore,

$$P(\mathrm{Vol}(A_{k_0,n}(X,\theta)\backslash D_X) > \varepsilon)$$
$$\leq P\left(\mathrm{Vol}(A_{k_0,n}(X,\theta)\backslash A_{k_0}^*(X,\theta)) > \frac{\varepsilon}{2} \text{ or } \mathrm{Vol}(A_{k_0}^*(X,\theta)\backslash D_X) > \frac{\varepsilon}{2}\right)$$
$$< \frac{\delta}{2} + \frac{\delta}{2} = \delta.$$

Finally, since the volume only gets smaller for more cuts than $k_0$, the result follows. $\qquad\square$

## B.2 Proof of the Main Theorem (Theorem 4)

We follow the same general outline as [32], with some notable exceptions. There they used the methods in [15], which provides tools for proving consistency results for estimators that minimize mean square loss among a data-dependent classes of functions. Note that both mean regression RFs and QRFs fall into this category.

Let $\mathcal{F}_n$ be the set of cell-wise constant functions on the (data dependent) partition given by the generalized tree from Theorem 4, and let $\mathcal{I}_n$ be the (random) set of indices of the subsample chosen.

We will now recast Theorem 3 from [32] (see also [15]), which was originally about mean regression, to the situation of using a generalized random forest estimator for quantile regression: (As $P(Y \leq y|X)$ is bounded, we may omit the truncation operators appearing in the original statements.)

**Theorem 5.** *Let $\mathcal{F}_n, \mathcal{I}_n$ be as above. Assume that there is a sequence of real numbers $\beta_n$ such that:*

1. *$\lim_{n\to\infty} \beta_n = \infty$.*

2. *The approximation error goes to $0$:*
$$\lim_{n\to\infty} \mathbb{E}\left[\inf_{g\in\mathcal{F}_n, ||g||_\infty \leq \beta_n} \mathbb{E}_X((g(X) - P(Y \leq y|X))^2)\right] = 0.$$

3. *The estimation error goes to $0$, namely for all $L > 0$:*
$$\lim_{n\to\infty} \mathbb{E}\left[\sup_{g\in\mathcal{F}_n, ||g||_\infty \leq \beta_n} \left|\frac{1}{a_n}\sum_{i\in\mathcal{I}_n}(g(X_i) - T_L I_{Y_i \leq y})^2 - \mathbb{E}((g(X) - T_L I_{Y \leq y})^2)\right|\right] = 0,$$
   *where $T_L(u) := \mathrm{sign}(u)\min(|u|, L)$.*

*then*
$$\lim_{n\to\infty} \mathbb{E}(|\hat{\eta}_1(X) - P(Y \leq y|X)|^2) = 0.$$

*Further, if the approximation and estimation errors converge uniformly in $y$, then so does $\mathbb{E}(|\hat{\eta}_1(X) - P(Y \leq y|X)|^2)$.*

We will now show that under the assumptions of Theorem 4 the conditions of Theorem 5 hold for $\beta_n = (\ln(\frac{a_n}{t_n \ln(a_n)}))^{\frac{1}{4}}$; and so Theorem 4 would follow.

### B.2.1 The approximation error goes to $0$ (uniformly in $y$)

Let $h(X) = \mathbb{E}(P(Y' \leq y|X')|X) \in \mathcal{F}_n$, where $(X', Y')$ is an IID copy of $(X, Y)$ restricted to the sub probability space where $X' \in R_{X,n}$. We see that

$$\mathbb{E}\left[\inf_{g\in\mathcal{F}_n, ||g||_\infty \leq \beta_n} \mathbb{E}_X((g(X) - P(Y \leq y|X))^2)\right] \leq \mathbb{E}(\mathbb{E}_X((h(X) - P(Y \leq y|X))^2))$$
$$= \mathbb{E}((h(X) - P(Y \leq y|X))^2)$$

As all of these quantities are bounded, it suffices to show that $|h(X) - P(Y \leq y|X)|$ goes to $0$ in probability. Assume by contradiction that there exists an $\varepsilon > 0$ for which there exists an $\varepsilon' > 0$ satisfying $P(|P(Y \leq y|X) - h(X)| > \varepsilon) > \varepsilon'$ for all $n$ big enough. Assumption 1b in particular implies that for every $\varepsilon'' > 0$ there exists a $\delta > 0$ so that $P(X' \in B_{\varepsilon''}(X)|X) > \delta$. ($P(X' \in B_{\varepsilon''}(X)|X) = P(X'' \in B_{\varepsilon''}(X)|X, X'' \in R_{X,n})$ for an IID copy $X''$ of $X$, which in

turn is at least $P(X'' \in B_{\varepsilon''}(X) \cap R_{X,n}|X)$; and since $B_{\varepsilon''}(X) \cap R_{X,n}$ with probability 1 includes a ball of radius at least $c\varepsilon''$ it follows that there exists a $\delta > 0$ satisfying $P(X' \in B_{\varepsilon''}(X)|X) > \delta$.) In particular $P(X' \in B_{\varepsilon''}(X)||P(Y \leq y|X) - h(X)| > \varepsilon) > \delta$. Using Assumption 1a choose $\varepsilon''$ so that $|P(Y' \leq y|X') - P(Y \leq y|X)| < \frac{\varepsilon'}{2}$ a.s. given $X' \in B_{\varepsilon''}(X)$. Then we have:

$$P(|P(Y' \leq y|X') - h(X)| > \varepsilon)$$
$$\geq \varepsilon'\delta P\left[|P(Y' \leq y|X') - h(X)| > \varepsilon \middle| |P(Y \leq y|X) - h(X)| > \varepsilon, X' \in B_{\varepsilon''}(X)\right]$$

By our choice of $\varepsilon''$, we have that given $|P(Y \leq y|X) - h(X)| > \varepsilon$ and $X' \in B_{\varepsilon''}(X)$ the following holds:

$$|P(Y' \leq y|X) - h(X)| \geq ||P(Y \leq y|X) - h(X)| - |P(Y' \leq y|X') - P(Y \leq y|X)|| \geq \frac{\varepsilon'}{2}$$

Thus, $P(|P(Y' \leq y|X') - h(X)| > \varepsilon) \geq \frac{\varepsilon'^2\delta}{2}$, and so does not converge to 0, in contradiction to the assumption that $\mathbb{E}(V(P(Y' \leq y|X')|X)) \to 0$ uniformly in $y$.

### B.2.2 The estimation error goes to $0$ (uniformly in $y$)

This follows the same pattern as in the proof of Theorem 1 in [32], except that in our case we can be more lax about our choice of $\beta_n$, because we don't have to concern ourselves with the untruncated situation.

Briefly, as in [32], by Theorems 9.1 and 9.4, and Lemma 13.1 in [15], as well as Assumption 2 in Theorem 4, we have that for all $\varepsilon > 0$, $\varepsilon' > 0$, and $L > 0$:

$$P\left[\sup_{g \in \mathcal{F}_n(\theta), ||g||_\infty \leq \beta_n} \left|\frac{1}{a_n}\sum_{i \in \mathcal{I}_{n,\theta}} (g(X_i) - T_L I_{Y_i \leq y})^2 - \mathbb{E}((g(X) - T_L I_{Y \leq y})^2)\right| > \varepsilon\right]$$

$$\leq \mathbb{E}(8\exp(-\frac{a_n}{\beta_n^4}C_n)|\frac{t_n\ln(a_n)}{a_n} < \varepsilon') + d_{\varepsilon',n};$$

$$\text{where } C_n = \frac{\varepsilon^2}{2048} - \frac{\beta_n^4 t_n \ln(da_n)}{a_n} - \frac{2\beta_n^4 t_n}{a_n}\ln(\frac{333e\beta_n^2}{\varepsilon}),$$

$d_{\varepsilon',n}$ is some sequence of non-negative numbers depending on $\varepsilon'$ satisfying that $\lim_{n\to\infty} d_{\varepsilon',n} = 0$, and where the randomness comes from the randomness of $t_n$. By our choice of $\beta_n$ and the use of L'Hôpital's rule, we get that $C_n$ converges to $\frac{\varepsilon^2}{2048}$ (uniformly in $y$, of course) in probability as $\varepsilon'$ goes to 0. To be precise:

$$\forall \varepsilon'' > 0 \lim_{\varepsilon' \to 0} P\left(|C_n - \frac{\varepsilon^2}{2048}| > \varepsilon'' \middle| \frac{t_n\ln(a_n)}{a_n} < \varepsilon'\right) = 0.$$

Following the steps of the remainder of the argument in [32], mutatis mutandis, this property suffices for the estimation error to go to 0. We remark that in [32] the number of cells $t_n$ is not a random variable, which makes the computation there somewhat more straightforward.

## C Conformalized Predictions

In what follows, let $(X_1, Y_1), ..., (X_n, Y_n) \in \mathbb{R}^d \times \mathbb{R}$ be the training set, and $(X_{n+1}, Y_{n+1})$ be a feature vector being queried at inference and its associated true value.

The classical conformalized prediction algorithm transforms a point prediction algorithm into an algorithm that outputs prediction intervals. To be precise, given a confidence level $\alpha$, we say that an algorithm is an ideal prediction interval forecaster if it takes $(X_1, Y_1), ..., (X_n, Y_n), X_{n+1}$ and outputs an interval $C$ satisfying that

$$P(Y_{n+1} \in C|X_{n+1}) = 1 - \alpha.$$

The conformalized prediction algorithm relaxes this requirement in two ways. First, it relaxes the equality to an inequality, and secondly it asks that the inequality would hold without conditioning (which roughly translates to the inequality holding on average on $X_{n+1}$):

$$P(Y_{n+1} \in C) \geq 1 - \alpha$$

It is this last condition that is guaranteed under exchangability assumptions on the data, and no assurances are made on conditional coverage.

The conformalized predictions algorithm is defined as follows: split the training data into two $\mathcal{I}_1$ and $\mathcal{I}_2$. Train a mean-regression point forecasting algorithm on $\mathcal{I}_1$, and let $\hat{\mu}$ be the resulting model.

The conformalized prediction interval for a new datapoint $X_{n+1}$ is then defined as the bootstrapping interval:

$$C := [\hat{\mu}(X_{n+1}) - Q_{1-\alpha}(\mathcal{I}_2), \hat{\mu}(X_{n+1}) + Q_{1-\alpha}(\mathcal{I}_2)],$$

where

$$Q_{1-\alpha}(\mathcal{I}_2) := (1 - \alpha)(1 + 1/|\mathcal{I}_2|)\text{-th emperical quantile of } \{|Y - \hat{\mu}(X_i)|\,\big|\,i \in \mathcal{I}_2\}.$$

We remark while this classical conformalized predictions algorithm behaves similarly to LSF in that it takes a point forecasting algorithm and output a probabilistic forecasting algorithm, the basic idea behind conformalized predictions is fundamentally different. Namely, conformalized predictions begin with the premise that an ideal prediction interval forecaster is too much to ask for, and instead of directly optimizing for it, it attempts to optimize instead the much weaker unconditional coverage requirement. Some algorithms, such as Conformalized Quantile Regression ([29]), attempt to straddle the two approaches: it takes two quantile regression point forecasters (rather than a single mean-regression forecaster), and then adjusts them to optimize for the unconditional coverage requirement, perhaps arriving at some regularizing effect. But they no longer serve as algorithms that turns point forecasting algorithms into probabilistic forecasting algorithms.

# D  Tree-based Time Series Wrapper (TTSW)

TTSW is implemented in GluonTS under the name Rotbaum: `https://github.com/awslabs/gluon-ts/blob/master/src/gluonts/model/rotbaum/`

---

**Algorithm 3:** Tree-based Time Series Wrapper (TTSW)

---

**Input**: Datasets made of multiple time series, where we denote the $i^{th}$ time series by $\{Z_{i,t}\}_t$; a context window size $h$; a forecast horizon $l$; and number of context windows to sample $P$.
`train_data, target_data, model_list = [], [], []`
**for** _ *in [1,...,P]* **do**
    Choose a time series $i$ and a beginning point within the time series $t_0$ uniformly. Add the context window $Z_{i,t_0}, Z_{i,t_0+1}, ..., Z_{i,t_0+h-1}$ to `train_data`. Add $Z_{i,t_0+h}, Z_{i,t_0+1}, ..., Z_{i,t_0+h+l-1}$ to `target_data`.
**for** *i in [1,...,l]* **do**
    Train a model (XLSF/Quantile Regression/QRF) with training data `train_data` and target data `target_data[:, i]`, and add it to `model_list`.

---

**Inference**: Given a new time series for inference $Z_1, ..., Z_k$, make inferences for the feature vector $(Z_{k-h+1}, ..., Z_k)$ from all of the models in `model_list`.

---

# E  Detailed TTSW Results

Table 4: Time Series Benchmarking

| | TTSW (XLSF) | | | TTSW (QuantileReg) | | | TTSW (QRF) | | | DeepAR | | | CNN-QR | | |
|---|---|---|---|---|---|---|---|---|---|---|---|---|---|---|---|
| | P10 | P50 | P90 | P10 | P50 | P90 | P10 | P50 | P90 | P10 | P50 | P90 | P10 | P50 | P90 |
| electricity | 0.0395 | 0.0726 | 0.0375 | 0.1203 | 0.2399 | 0.0833 | 0.0624 | 0.1190 | 0.0541 | 0.0497 | 0.0929 | 0.035 | 0.0476 | 0.0918 | 0.0408 |
| parts | 0.4371 | 1.3362 | 1.1396 | 0.2 | 1.017 | 1.0582 | 0.2826 | 1.3491 | 1.2536 | 0.2225 | 1.0135 | 1.7272 | 0.2028 | 1.0048 | 1.0623 |
| m4_daily | 0.013 | 0.0204 | 0.0112 | 0.0385 | 0.0281 | 0.018 | 0.0157 | 0.0263 | 0.0146 | 0.0217 | 0.0354 | 0.0149 | 0.0144 | 0.0209 | 0.0112 |
| traffic | 0.0797 | 0.1682 | 0.1158 | 0.0709 | 0.1406 | 0.1072 | 0.0752 | 0.1707 | 0.141 | 0.057 | 0.1307 | 0.0864 | 0.0641 | 0.1578 | 0.1186 |
| wiki10k | 0.2331 | 0.3573 | 0.2874 | 0.1696 | 0.3127 | 0.2782 | 0.1801 | 0.3571 | 0.3835 | 0.1458 | 0.3062 | 0.2743 | 0.1732 | 0.3048 | 0.2668 |
| dcrideshare | 0.2028 | 0.5289 | 0.2829 | 0.1634 | 0.4887 | 0.2994 | 0.2148 | 0.6006 | 0.3636 | 0.1846 | 0.4318 | 0.2348 | 0.1764 | 0.4422 | 0.2277 |

# F  Choices of Hyperparameters

For the purpose of using Algorithm 3 for applying XLSF to time-series data, we always use `min_bin_size=` 100, and the following hyperparameters for the underlying XGBoost model:

```
XGBModel(base_score=None, booster=None, colsample_bylevel=None,
        colsample_bynode=None, colsample_bytree=None, gamma=None, gpu_id=None,
        importance_type='gain', interaction_constraints=None,
        learning_rate=None, max_delta_step=None, max_depth=5,
        min_child_weight=None, missing=nan, monotone_constraints=None,
        n_estimators=100, n_jobs=-1, num_parallel_tree=None,
        objective='reg:squarederror', random_state=None, reg_alpha=None,
        reg_lambda=None, scale_pos_weight=None, subsample=None,
        tree_method=None, validate_parameters=None, verbosity=1)
```

We use a context length $h$ equal to the forecast horizon $l$, and we set the number of context windows $P$ to equal 1000000, unless the dataset is small, in which case it automatically chooses less. Since QRF is much slower to train, we had to reduce $P$ to 10000, which means it had 100 times less data-points to train on. Even with this adjustment, it took considerably longer than the other methods.

In Section 6.1, we use `min_bin_size=` 100, and (since the resulting datasets are much smaller than the ones used in time series prediction) the following hyperparameters for the underlying XGBoost model:

```
XGBModel(base_score=0.5, booster='gbtree', colsample_bylevel=1,
        colsample_bynode=1, colsample_bytree=1, gamma=0, gpu_id=-1,
        importance_type='gain', interaction_constraints='',
        learning_rate=0.300000012, max_delta_step=0, max_depth=2,
        min_child_weight=1, monotone_constraints='()',
        n_estimators=100, n_jobs=-1, num_parallel_tree=1,
        objective='reg:squarederror', random_state=0, reg_alpha=0,
        reg_lambda=1, scale_pos_weight=1, subsample=1, tree_method='exact',
        validate_parameters=1, verbosity=1)
```

For the purpose of the M5 competition, we use `min_bin_size=` 200, since we query half-percent quantiles.

For both QRFs and lightgbm quantile regression we use default hyperparameters:

```
RandomForestQuantileRegressor(bootstrap=True, criterion='mse', max_depth=None,
                            max_features='auto', max_leafodes=None,
                            min_samples_leaf=1, min_samples_split=2,
                            min_weight_fraction_leaf=0.0, n_estimators=10,
                            n_jobs=1, oob_score=False, random_state=None,
                            verbose=0, warm_start=False)
```

and

```
LGBMRegressor(boosting_type='gbdt', class_weight=None, colsample_bytree=1.0,
              importance_type='split', learning_rate=0.1, max_depth=-1,
              min_child_samples=20, min_child_weight=0.001, min_split_gain=0.0,
              n_estimators=100, n_jobs=-1, num_leaves=31, objective=None,
              random_state=None, reg_alpha=0.0, reg_lambda=0.0, silent=True,
              subsample=1.0, subsample_for_bin=200000, subsample_freq=0)
```

# G   Detailed Tabular Data Experiments

We have re-run the tabular experiments from Section 6.1 five times to get confidence intervals. Note that XLSF is deterministic because the hyperparameters we chose for XGBoost (see Appendix F) preclude subsampling.

| Algorithm | Dataset | P05 | P95 | accuracy | time (s) |
|---|---|---|---|---|---|
| XLSF | $facebook_1$ | **0.103+/-0.0** | **0.288+/-0.0** | 94.31%+/-0.0% | 7.801+/-0.022 |
| | $facebook_2$ | 0.097+/-0.0 | 0.294+/-0.0 | 95.327%+/-0.0% | 14.883+/-0.051 |
| | $meps_{19}$ | **0.101+/-0.0** | **0.563+/-0.0** | 93.38%+/-0.0% | 6.285+/-0.335 |
| | $meps_{20}$ | **0.101+/-0.0** | **0.664+/-0.0** | 92.79%+/-0.0% | 5.926+/-0.037 |
| | $meps_{21}$ | **0.101+/-0.0** | **0.556+/-0.0** | 92.688%+/-0.0% | 6.386+/-0.132 |
| | concrete | 0.037+/-0.0 | 0.039+/-0.0 | 76.699%+/-0.0% | 0.712+/-0.005 |
| | star | 0.012+/-0.0 | **0.012+/-0.0** | 78.522%+/-0.0% | 1.108+/-0.001 |
| | bio | 0.082+/-0.0 | 0.132+/-0.0 | 87.634%+/-0.0% | 13.648+/-0.295 |
| | community | 0.106+/-0.0 | 0.19+/-0.0 | 76.19%+/-0.0% | 1.186+/-0.011 |
| | bike | 0.057+/-0.0 | 0.06+/-0.0 | 87.741%+/-0.0% | 3.452+/-0.02 |
| QRF | $facebook_1$ | **0.103+/-0.005** | 0.319+/-0.013 | 92.383%+/-0.203% | 30.368+/-0.121 |
| | $facebook_2$ | **0.094+/-0.001** | **0.289+/-0.006** | 92.918%+/-0.147% | 104.485+/-2.923 |
| | $meps_{19}$ | 0.11+/-0.002 | 0.727+/-0.018 | 89.914%+/-0.516% | 9.798+/-0.314 |
| | $meps_{20}$ | 0.107+/-0.001 | 0.687+/-0.028 | 88.663%+/-0.331% | 10.623+/-0.043 |
| | $meps_{21}$ | 0.108+/-0.001 | 0.655+/-0.024 | 88.57%+/-0.371% | 9.246+/-0.095 |
| | concrete | 0.036+/-0.002 | **0.037+/-0.001** | 80.194%+/-2.109% | 0.175+/-0.001 |
| | star | 0.013+/-0.0 | 0.014+/-0.001 | 79.723%+/-0.853% | 0.518+/-0.002 |
| | bio | **0.072+/-0.0** | **0.098+/-0.001** | 84.682%+/-0.274% | 33.258+/-0.283 |
| | community | **0.088+/-0.005** | **0.168+/-0.007** | 87.569%+/-1.125% | 1.14+/-0.008 |
| | bike | **0.044+/-0.001** | **0.046+/-0.001** | 80.358%+/-0.461% | 2.842+/-0.036 |
| CP | $facebook_1$ | 0.452+/-0.011 | 0.425+/-0.005 | 90.147%+/-0.151% | 7.028+/-0.049 |
| | $facebook_2$ | 0.43+/-0.019 | 0.435+/-0.007 | 90.22%+/-0.162% | 14.552+/-0.077 |
| | $meps_{19}$ | 0.657+/-0.039 | 0.76+/-0.011 | 90.459%+/-0.455% | 5.423+/-0.109 |
| | $meps_{20}$ | 0.488+/-0.026 | 0.774+/-0.014 | 90.208%+/-0.166% | 5.926+/-0.037 |
| | $meps_{21}$ | 0.504+/-0.034 | 0.8+/-0.007 | 90.134%+/-0.711% | 5.288+/-0.039 |
| | concrete | **0.034+/-0.002** | 0.038+/-0.002 | 87.767%+/-3.482% | 0.091+/-0.0 |
| | star | **0.011+/-0.0** | **0.012+/-0.0** | 88.037%+/-0.717% | 0.309+/-0.002 |
| | bio | 0.104+/-0.001 | 0.13+/-0.001 | 89.672%+/-0.309% | 4.561+/-0.039 |
| | community | 0.134+/-0.003 | 0.181+/-0.006 | 92.632%+/-1.192% | 0.959+/-0.007 |
| | bike | 0.05+/-0.001 | 0.049+/-0.0 | 90.101%+/-0.783% | 0.737+/-0.003 |

Table 5: Benchmarking results: the datasets were taken from `https://github.com/yromano/cqr/tree/master/datasets`. Accuracy is percent of times the true value was in the prediction interval. (It should revolve around 90%.) P05 and P95 are weighted quantile losses. The experiments were run 5 times.

## H Formula for Weighted Quantile Loss

The formula for weighted quantile loss for quantile $\tau$, for real values $y_i$, and predictions $q_i$:

$$wQL^{Reg}[\tau] := 2\frac{\sum_i \tau \max(y_i - q_i, 0) + (1 - \tau)\max(q_i - y_i, 0)}{\sum_i |y_i|}$$

In the time series situation, if $y_{i,t}$ as $t$ varies are the real values for time series number $i$ in the dataset, and $q_{i,t}$ is prediction for $y_{i,t}$ then the formula is:

$$wQL^{TS}[\tau] := 2\frac{\sum_{i,t} \tau \max(y_{i,t} - q_{i,t}, 0) + (1 - \tau)\max(q_{i,t} - y_{i,t}, 0)}{\sum_{i,t} |y_{i,t}|}$$

## I LSF with Non-Tabular Data

One of the assumptions of LSF is that the base point-forecasting algorithm is tabular, and so that raises the question of how to integrate LSF with neural networks dealing with non-tabular data. To that end, LSF can come in at the embedding level of the architecture. To be a little more explicit, if $f_1$ is the portion of the neural network that embeds the data into a fixed dimensional space, and the remainder of the network is $f_2$, then after training you would cache pairs of embeddings and true target values, and later feed them into LSF together with $f_2$ as the base algorithm. At inference, rather than feeding in the raw data, you would feed the data into $f_1$ to obtain a fixed length feature vector, and only then feed it into the LSF.