# OpenReview forum: "Probabilistic Forecasting: A Level-Set Approach"
_NeurIPS.cc/2021/Conference — NeurIPS 2021 Poster_

### Official Review · Reviewer_Bgd7 · 2021-07-12

**Rating:** 6
**Confidence:** 2

**Summary:**

This paper proposes a level-set approach (LSF) to model the uncertainty in forecasting problems. It can process the result of any point forecaster into probabilistic forecasts. The main idea of LSF is to group the predictions of training data that are sufficiently close, which is also used to partition the feature space. One then obtains different bins of true values in the training set, and this can be used as the predicted distributions. The authors have proved the mean-square consistency of LSF and further extended to random forests in more general settings. Empirical evaluations on both tabular and time series competition datasets demonstrate the effectiveness of this method.

**Limitations And Societal Impact:**

The authors have addressed social impacts but haven't discussed limitations. From the experimental section, the computational efficiency and coverage guarantee compared with conformal prediction approaches, as well as the generalization to sequential data with hierarchical aggregations might be good directions.

**Main Review:**

This paper has a clear motivation and is well structured. The LSF partitioning algorithm is theoretically sound in terms of the mean-square consistency. Prior works have shown the consistency of random forests. The authors further prove that LSF is a generalized QRF with tree-growing partitioning algorithms. Overall, LSF can produce probabilistic forecasts at any specified quantile, which is distribution-free, but I am not sure if the binning procedure can be implemented more efficiently.

In addition, since the authors mentioned (line 33) that LSF can turn any point estimator into a probabilistic one, I wonder if the results (both LSF algorithm and theory) can extend to point predictors other than XGBoost. In that case, is preprocess to tabular data still necessary?


**Time Spent Reviewing:**

6 hours

---

> ### Author Response · Authors · 2021-08-10
> **Response to Bgd7**
>
> Thank you for the thoughtful feedback, as well as for the kind words regarding the paper being well structured and with a clear motivation. In what follows, we answer the two major concerns that you have.
>
> 1. Whether LSF is implemented efficiently. In this regard we would merely like to emphasize that at the end of Algorithm 1, the dictionary taking target values in train to bins of true target values (pred_to_bin) satisfies that for keys that map to the same bin, this dictionary maps them to the same list object rather than to identical but equal list objects. This not only saves on memory, but at inference (as described in lines 83-85), the caching of the computation of the quantile of a particular bin can be recalled later if the bin is being referred to by using a different key of pred_to_bin. This little trick goes a long way in making the implementation fast and memory effective. We would say that the greatest bottleneck in terms of optimizing further is that for each new feature vector $x$ we do a binary search for which of $\{v_1,...,v_k\}$ is closest to $f(x)$. We would also like to emphasize that LSF applied to XGBoost is much faster than skgarden’s implementation of quantile regression forests, which is what allows us to use it for the time series forecasting use case, which requires large data to work well.
>
> 2. You asked whether LSF requires preprocessing into tabular data, or can it be applied more generally. The preprocessing into tabular data is necessary, but it can also be integrated into certain neural network architectures by caching the data at the embedding level during training. To give an example, say that you have a neural network model that takes documents to real values, where the documents are of varying length. Say further that the documents are first embedded into vectors of fixed length (call this portion of the neural network $f_1$), and that the remainder of the architecture is an MLP (call this portion $f_2$). In that case, during training you would cache pairs of embeddings and true target values, and later them feed them into LSF together with $f_2$ as the base algorithm. At inference, rather than feeding in the raw documents, you would feed the document into $f_1$ to obtain a fixed length feature vector, and then feed that into the LSF. We will raise this point in the final draft to encourage adoption for neural networks, and we thank you for encouraging us to clarify this further in the paper.

---

### Official Review · Reviewer_eTYD · 2021-07-14

**Rating:** 7
**Confidence:** 3

**Summary:**

The authors propose a general method to derive probabilistic / conditional distribution predictions for any given trained base prediction model f(x) - i.e., CDF of y|x, by using an approach somewhat related to quantile regression forests (QRFs).  In this approach prediction function f(x_i) outputs are grouped with nearby values, so that groups have a minimum size from the training data.  Each group's true values {y_i} define a sample for that group that can be used to compute any CDF value - and, for instance provide a prediction interval via a set of quantiles.  A prediction for a new sample x is transformed into the probabilistic prediction by finding the nearest group to f(x), and using its samples to derive the distribution statistics.

The authors prove consistency under fairly reasonable assumptions of the prediction algorithm, as well as proving consistency for QRFs with CART splitting.

The authors also perform extensive experiments showing the proposed approach is effective at estimating distribution statistics / prediction intervals for a variety of datasets, compared to other approaches to deriving distributions from point predictions (conformal approach) as well as other probabilistic approaches (QRF, quantile regression approaches, DeepAR), showing it is both efficient and competitive in terms of accuracy, even out-performing other methods.  These include experiments with forecasting datasets showing how their approach applied to the point forecasting winner of M5 made it competitive for probabilistic forecasting with the top 5 probabilistic forecasting methods from the competition.



**Limitations And Societal Impact:**

I feel more could be discussed around the limitations of the algorithm, and practical implications of the assumptions.  In particular, in which kind of cases, for what kind of datas or learning algorithms, would the method be unlikely to work?

**Main Review:**

I found this to be a very interesting paper that is also very useful.

**Originality:** This seems like a pretty novel approach.  Although it has some similarity to QRF, it is a generic procedure applicable to any model, and much different to decide the groups representing the conditional distributions.  Unlike QRF that is based on a specific learning and prediction algorithm only, and much slower.   Also it includes novel proofs and experiments.

**Significance:** This work seems very significant and useful.  The majority of models are point prediction models (and these are often easier and more efficient to create and train), but for many reasons its very desirable to have prediction intervals and distributional predictions - e.g., for making decisions based on predictions.  So having some procedure like this that is effective and efficient at transforming point predictions into probabilistic ones is extremely useful, across many domains.  This is a step in that direction, which could be useful in practice and spawn new research in this direction as well.

**Clarity:**
Overall the paper is well written and clear - there are a few points that are hard to follow, when describing the approach / algorithm - it takes some careful reading through to fully understand.  It might be helpful here to have a simple example to illustrate the approach - as this would immediately make it clear, or some visual illustration of the approach.

**Quality:**
The approach and details seem technically sound (though I did not check the proofs carefully).  That being said, I think the work could definitely use more discussion around its limitations - including the practical implications of the assumptions and the approach, and under which cases it may not work well.  I can think of several cases where it wouldn't work, and also found this approach raised several questions about its viability (see below).  That being said, the thorough set of experiments does show that despite these assumptions and cases it may fail it is generally effective with a lot of real data and at least one common class of learning algorithms (gradient boosting) - but it is important for readers to have some idea of when it might fail.

It would also be ideal to show results when using it with other algorithms as well besides just gradient boosting (such as deep learning models, linear models, SVM, random forest, etc.).

Also, On the claims:
One claim is for the proof of consistency of QRFs,  however consistency was already proven in [21] for QRFs – so what is the specific differentiating claim here?  Later text seems to make clear it is for a different splitting criteria than previously proven.  This should be made clear  up-front that the differentiating contribution is here and that consistency was already proven in the prior work.


Here are some detailed questions and potential limitations that could use more discussion, and other issues:


**1)	Practical implications of assumptions:**
More should be discussed about the practical implications of the assumptions and when this works or when this doesn’t, based on the algorithm A and the data properties.

For example, how can this not depend on consistency of the algorithm A, or to converge to it predicting some particular statistic?  Clearly this is interestingly tied to the assumptions, such as Assumption 2a.
E.g., in an extreme case, if A is an algorithm that returns a function always predicting the same value, then nothing can be gained with this approach.  In a more reasonable case, what if A always predicts a median of a sparse count distribution (like Poisson with mean << 1).  Again it will always predict 0, despite being completely accurate in estimating the median, and again the correct quantiles of the underlying distribution cannot be obtained with this method.  There could be cases in between these extreme cases, e.g., estimating the median but still having a good proportion of values > 0.

Another related question along these lines, if the point prediction is for some other statistic, say itself target at the 0.1 quantile, then does this method still return the conditional CDF of y given x?  I.e., if f(x) is trained to predict quantile 0.1, e.g., with quantile loss.

I also wonder what the implications are for the consistency of algorithm A itself, as this seems to suggest that algorithm A could basically be anything and need not be consistent.  Further, if the underlying data is stochastic, and algorithm A can at best predict a statistic – does this still hold?  Some experiments that would be good to show include using different algorithms as the base algorithm including ones that underfit (like linear models), and also even arbitrary functions.

In general, some interpretation and discussion of the implications of the assumptions, and analyses and experimentation around when the method works well or not, would be very helpful to include.


**2) Heteroskedasticity:**
Another major concern is this approach seems to impose strong assumptions, that if the prediction function output values are similar, the underlying distribution is also similar, but in general this may not be the case and could lead to incorrect results, as it doesn’t depend on any grouping of the data (x_i) itself, only f(x_i).  In fact it’s fairly common for time series from real world applications to exhibit heteroskedasticity.  For example, going through time periods of cyclical or event-driven increased volatility/uncertainty.  In other words, for one set of inputs and/or at one point in time the mean (and correspondingly the prediction, f(x), of a good prediction model) could be exactly the same as for another set of inputs and/or another point in time, but the variance much different, and the underlying distribution much different (and thus the prediction interval much wider for one than the other).  But because the predicted mean value is the same, they will be grouped into the same group, and the prediction intervals estimated for both time period will be the same, and incorrect.

E.g., consider commodity prices, right before government or other major reports are released, the mean prediction for the change in price might be the same as recent mean predictions, but the variance in that prediction should be much wider, as it depends largely on the yet unforeseen news.  In practice, the true value for the next time period will very much more than during other stable times.


**3) Overfitting:**
I wonder if the authors could comment on the case of models that typically overfit like neural nets.  These days in many cases neural nets will achieve 0 training error before the test error reaches its minimum.  This method would not seem to work at all for such models – which makes me question its utility when applied to modern DL algorithms.

-->Again it would be good to see results with more algorithms besides just gradient boosting…

**4)	Definitions of the metrics used are missing**
In particular quantile loss should be defined, at least in the appendix – and also explained what weights are used as Table 1 says “weighted quantile loss” but weights are never explained



**Minor comments:**

As mentioned, another option for forecasts is to output a vector of results – i.e., f(x) is a vector  - how would you apply the approach in this case?


Line 72-72 why is k not equal to n?  Should add an explanation here, that {v_j} is the set of unique values of {f(x_i)} – I’m assuming this is what is meant but it should be spelled out.  This becomes clear later but would make it easier to follow if this is explained when first introduced.

In Algorithm 1 – is the sufficient bin size based on all values of y_i (even if they are the same value)?  It seems to be so but it might help to note this in the text.



**--Update after response and discussion--**

I appreciate the authors' responses and addressing the many reviewer comments - I think it will make it an even stronger paper.  Kept my rating, and looking forward to reading the final version!




**Time Spent Reviewing:**

3

---

> ### Author Response · Authors · 2021-08-10
> **Response to eTYD**
>
> Thank you for the thoughtful feedback, as well as for the kind words regarding the novelty of the paper, its usefulness, and the thoroughness of the experiments. We too hope that it will spawn new research in this direction.
>
> You asked several questions regarding the assumptions and their relationship to homoscedasticity of the data, consistency of the base algorithm, the statistic which the base algorithm is regressing, and under/over fitting of the base algorithm. Our response to these questions has some overlap with our response to Reviewer Zoe4, albeit with a different emphasis.
>
> We agree that the assumptions in the manuscript are somewhat restrictive. Since the submission, we have managed to simplify and generalize them, which will facilitate an easier discussion regarding the limitations of the method. The main limitation of the method is that it will fail if the conditional mean is constant over some subset of feature space, but on that same subset the conditional distribution is not constant (because the base algorithm regresses the conditional mean). In order to rule this out, we will have to assume, as you say, that “if the prediction function output values are similar, the underlying distribution is also similar”; or, breaking it into two: 1. Data Assumption: If the conditional mean is similar then the underlying conditional distribution is similar (change in the conditional CDFs can be bounded by any $\varepsilon>0$ given a sufficiently small change in the conditional mean), and 2. Base Estimator Consistency Assumption: The base estimator is mean-square consistent. Once we replace the current data assumption, Assumption 1, with this Data Assumption above, we are able to replace Assumption 2.d in the manuscript, the least intuitive and most restrictive of the assumptions, with the Base Estimator Consistency Assumption, which is more intuitive and easier to satisfy.
>
> With this framing, your questions can be address as follows: the Data Assumption above does not impose homoscedasticity. The conditional variance is not allowed to vary only for a fixed conditional mean, but it is allowed to vary as the conditional mean changes. If the conditional variance varies wildly as the conditional mean changes, it has implications on the rate of convergence of the consistency of LSF, not about the consistency itself. It is true that under/over fitting of the base algorithms will in general preclude consistency guarantees of LSF: in other words, it is only guaranteed to be as good as its base point prediction algorithm, though it is possible to come up with special examples where it would be robust to under/over fitting. In preparation of our manuscript, we experimented with linear regression as the base algorithm to confirm our intuition that it performs less well than XGBoost as the base algorithm. Based on these (expected) outcomes, we did not further elaborate on this in the paper. Finally, if you replace the base algorithm with a regressor of another statistic, say the 0.1 conditional quantile, everything transfers. But you will have to change the Data Assumption above mutatis mutandis to be about the 0.1 conditional quantile.
>
> As you have remarked, the experiments show that the algorithm is effective with real world data, and so the Data Assumption above can be seen as preventing a degenerate synthetic counter-example that is not likely to occur naturally. As we have mentioned to Reviewer Zoe4, Assumptions 2.a, 2.b and 2.c are all intuitive in that they heuristically only require that $f$ spreads feature space in a sufficiently non-degenerate manner. For example, they trivially hold if the base algorithm returns a uniformly random number between $0$ and $1$ with no regard to the feature vector; but all fail if the base algorithm is constant. We will include all of these details in the final draft.
>
> We now address the more minor questions. You asked if Meinshausen's "Quantile regression forests" has not already proven consistency of QRFs. Yes, but under the unrealistic regime of label-independent splitting criteria, unlike the CART-splitting criterion. We will make this distinction explicit in the final draft.
>
> We agree that the definition of weighted quantile loss should be added (this was also requested by Reviewer Zoe4).
>
> You asked about a possible generalization where the $y_i$'s are vectors. In that case it is a little unclear what probabilistic predictions should mean. Interesting work in this direction is Guillaume Carlier, Victor Chernozhukov, and Alfred Galichon's paper "Vector quantile regression:  an optimal transport approach", but for this paper it is out of scope.
>
> You asked if in Algorithm 1 the sufficient bin size is for the values of the $y_i$'s is with repetition. Yes, we can make that more explicit. We can also be more explicit that the reason that $k$ is not equal to $n$ in lines 72-77 is, as you deduced, that $k$ is the number of unique values of the $f(x_i)$'s.

---

### Official Review · Reviewer_Zoe4 · 2021-07-31

**Rating:** 7
**Confidence:** 3

**Summary:**

The authors propose a method for turning any prediction model into a probabilistic forecaster. Briefly, the approach works by grouping training data with similar outputs, and then considers the binned true values as a prediction distribution. They prove that their algorithm is consistent by extending proofs about the consistency of random Forests to more general contexts. To evaluate their method, the authors analyze publicly available tabular datasets and time series datasets. They also adapt use their method to adapt the M5 forecasting competition’s winner to perform probabilistically, and compare their results to the top entrants in the M5 uncertainty competition.


**Ethical Concerns:**

The authors do not discuss limitations in detail, as I described in the previous section. It is difficult to imagine a specific negative societal impact, beyond whatever negative societal impact any forecasting method has. Impacts would be more associated with potential applications.


**Limitations And Societal Impact:**

The authors do not discuss limitations in detail, as I described in the previous section. It is difficult to imagine a specific negative societal impact, beyond whatever negative societal impact any forecasting method has. Impacts would be more associated with potential applications.


**Main Review:**

Originality- The method proposed to make probabilistic inferences from point wise forecasters is fairly intuitive, but the rigorous proofs and evaluations show that their method performs well and is theoretically sound. I do not know of any other work that turns point-wise forecasters into a probabilistic method with so much rigor. So overall it is a novel method.

Significance- Pointwise models are standard, but most forecast applications require only knowing where an output lies generally, ie knowing what range an output will be in is often as useful as exact estimates. Furthermore, explainability is a hot issue right now, and knowing a model’s confidence aids in that tremendously. Overall, a method that is able to take any high-quality forecasting method and utilize it to learn more interpretable models is quite useful.

Clarity- Many aspects of the paper were clear, for instance many of the high-level overviews were easy to follow and motivated the rest of the paper. The overall method was also easy to follow. Some of the implementation and evaluation details, particularly pertaining to the M5 competition and benchmark datasets, were difficult to follow, as were some of the assumptions.

Quality-Overall, this is a high quality paper, with a useful method, strong proofs and extensive evaluations. What is most limiting is a lack of details and clarity in the evaluation section and in some other areas, which I will describe below.



Issues-

1) Algorithm 1- by always merging with the bin to the right (except at the end), it is possible to wind up with very lopsided bins. For instance, if your minimum bin size were 10, and your initial bins were: 10,2,3,5,40, you would wind up with two bins of size 10 and 50, where as if you merged to the left, they would be 20 and 40, which is much closer. Is there a reason why this doesn’t pose a big issue?

2) Assumptions- these could be elucidated with some definitions-  for instance, what does volume mean in this setting, and the description of collisions is unclear. Overall, some high-level descriptions of the assumptions, along with a discussion regarding how likely they are to hold, would be quite useful.

3) Overall evaluation- the main evaluation metric, weighted quantile loss, is not described anywhere. Also it is a little confusing that each table has different metrics and methods, although that is a minor issue. It is also not clear if XLSF is better compared to QRF in table 1- they seem to outperform each-other an equal amount across the different datasets and metrics.

4) M5 data analysis- The authors only consider the bottom level of the hierarchy, but in the competition, they checked both. Are the results for the competitors in table 2 what was reported in the M5 competition, or the author’s updated metric? If they did not get the same metric for both datasets, that does not appear to be a fair comparison, otherwise they should clarify that the metric is the same for all methods.

5) Benchmark datasets- where did these datasets come from? I understand that they are commonly available, but there are no references or descriptions that I could find. It also appears that their model is outperformed by deep baselines for most datasets, is this not a limitation?

6) Limitations- the authors do not discuss many limitations. What happens in the case where any assumptions are violated? How likely is it that any of these assumptions would be violated? Building a model for each timepoint doesn’t take advantage of shared information between timepoints like a multi-step model does. Does this method generalize to small datasets, or skewed datasets? Any comments amount where and why it was outperformed?

**Time Spent Reviewing:**

3

---

> ### Author Response · Authors · 2021-08-10
> **Response to Zoe4**
>
> Thank you for the thoughtful feedback, as well as for the kind words regarding the overall quality of the paper, its novelty, its usefulness, and its significance.
>
> You have cited some lack of clarity in the evaluation section as the main limiting factor of an otherwise high quality paper. Thankfully, this is something we can easily clarify. For the M5 competition, we are indeed using the same metric for all methods in the table, namely weighted quantile loss only for the bottom level of the hierarchy, and so it is a fair comparison. We will amend our article with citations for the datasets, as well as adding formulas for weighted quantile loss (asked by Reviewer eTYD as well).
>
> It is true that XLSF (LSF applied to XGBoost) does not exhibit clearly superior performance over QRF, but it also performs a different function: LSF turns a general point prediction algorithm into a probabilistic one, and XLSF is merely one application that produces a tree-based probabilistic algorithm. All we set out to show in that comparison is that it matches the performance of the main alternative tree-based probabilistic algorithm. Additionally, our implementation of XLSF is much faster than skgarden's implementation of QRF, which allows for it to scale to larger datasets.
>
> You are correct that merging the bins to the right, rather than the left, is arbitrary, but asymptotically this choice will not matter since the number of bins and the minimal bin size are both controlled (by letting min_bin_size be the order of magnitude of $(\ln(n))^2$), and since the image of the feature vectors in the training data under the base model are assumed to be dense in probability. We will add this discussion to the paper.
>
> You asked if building a model for each timepoint in the future fails to take advantage of the shared information between timepoints like a multi-step model does. We assume that by multi-step model you meant one model predicting the entirety of the forecast horizon. We have not witnessed this being an issue for time series models in general: DeepAR, which we consider to have state-of-the-art performance, is autoregressive and so is not multi-step. On the other hand, another of the state-of-the-art forecasters, Multi-Horizon Quantile Forecast (MQ-CNN) (Wen et al., 2017), which is multi-step, treats each forecast horizon independently in the decoder part, effectively building one quantile decoder at each timepoint. We remark also that our experiment section includes both small data (the tabular data in these experiments is all small) and large data (in the time series use case).
>
> Finally, we agree that the assumptions in the manuscript are somewhat restrictive. Since the submission, we have managed to simplify and generalize them, which will facilitate an easier discussion regarding the limitations of the method. The main limitation of the method is that it will fail if the conditional mean is constant over some subset of feature space, but on that same subset the conditional distribution is not constant (because the base algorithm regresses the conditional mean). Note that, as Reviewer eTYD has remarked, the experiments show that the algorithm is effective with real world data, and so this restriction can be seen as a degenerate synthetic counter-example. In our simplified assumptions, we rule this situation out by replacing Assumption 1 with an assumption that says that change in the conditional CDFs can be bounded by any $\varepsilon>0$ given a sufficiently small change in the conditional mean. Assumption 2.d, the least intuitive and most restrictive of the assumptions, can then be replaced with the more intuitive and easier to satisfy assumption that the base algorithm is (mean square) consistent. Assumptions 2.a, 2.b and 2.c are all intuitive in that they heuristically only require that $f$ spreads feature space in a sufficiently non-degenerate manner. For example, they trivially hold if the base algorithm returns a uniformly random number between $0$ and $1$ with no regard to the feature vector; but all fail if the base algorithm is constant. We will include these details and their discussion in the final draft.

---

> ### Comment · Reviewer_Zoe4 · 2021-09-10
> **Response to Authors**
>
> The authors have responded to my concerns, and I've increased my score. Great work by the authors.

---

### Author Response · Authors · 2021-08-26
**Thank you to the reviewers and chairs**

We would like to thank the reviewers again for the time they dedicated to reviewing our work and for their insightful comments. As the discussion period is almost over, we hope that we have adequately answered all the reviewers' comments. If further clarifications are needed, please let us know.

---

### Decision · Program_Chairs · 2021-09-27

**Decision:**

Accept (Poster)

**Comment:**

This work proposes a novel approach for transforming point estimators into probabilistic estimators. Overall, reviewers appreciate the novelty and significance of the work. However, they also made several important recommendations regarding the presentation and specifically the discussion of limitations. The authors should carefully consider reviewer feedback when working on their revision.